# Assessing the Role of Extreme Mediterranean Events on Coastal River Outlet Dynamics

Florian Meslard [1], Yann Balouin [2], Nicolas Robin [1] and François Bourrin [1,*]

1    Centre de Formation et de Recherche sur les Environnements Méditerranéens, Université de Perpignan Via Domitia, UMR 5110, 52 Avenue Paul Alduy, CEDEX, 66860 Perpignan, France
2    Bureau de Recherches Géologiques et Minières, Université de Montpellier, 1039 Rue de Pinville, 34000 Montpellier, France
*    Correspondence: fbourrin@univ-perp.fr

**Abstract:** River mouths are highly dynamic environments responding very rapidly to changes in wave energy or river floods. While the morphological response during floods or during marine storm events has been widely documented in the literature, little is known about the mechanisms acting during the co-occurrence of fluvial and marine hazards. This concomitance of river flood and marine storm is quite common in the western Mediterranean Sea, and was the case for the Gloria event, considered to be the most extreme event in recent decades. During this event, monitoring of hydrodynamics and morphological evolution was implemented, making it possible to better understand the impact of concomitant marine storm and fluvial flood during an extreme meteorological event on spit breaching of a small Mediterranean river mouth. Monitoring using a combination of high-resolution hydrodynamic measurements, topographic and bathymetric surveys, and sediment cores was used before, during, and after the storm "Gloria". The results suggest an amplification of the morphological impact of the events and a different morphogenic response than if each of the events had acted independently on the system. The marine storm, occurring first, weakened the spit and initiated its breaching, which was continued by the extreme fluvial flood, thus leading to the complete destruction of the mouth. The destruction of the spit acted as a sediment source for subaqueous large delta deposition amounting to 50% of the total volume. The contribution of the river, estimated at 30%, was quite low for an exceptional event, showing the importance of locating rainfall in a catchment area controlled by a dam. For this event, extreme morphological evolution was observed, as well as the importance of water levels in the river mouth, which probably increased flood hazards, demonstrating the importance of including the compounding effect of extreme coastal water levels in river flood risk management.

**Keywords:** extreme event; river mouth; spit breaching; concomitant marine storm and fluvial flood; IOCE

## 1. Introduction

Coastal river outlets are dynamic landforms at the transition between fluvial and marine environments [1–3]. They are important from both economic (protection of port and marina access, sediment reservoir, etc.) and environmental (fish habitats and migration, water quality, etc.) standpoints (e.g., [4–6]). Coastal river outlets can be observed in a wide variety of environments, ranging from micro–mesotidal (e.g., [7–9]) to macro–megatidal environments (e.g., [10–12]), and influence the morphodynamic behavior and sediment budgets of adjacent shorelines over several kilometers [13,14]. Their morphology and evolution are dependent on the complex interaction between river outflow, waves and tides acting over a broad spatial and temporal scales, and inherited outlet morphology or geological framework [15–17]. In wave-dominated microtidal systems with low or variable fluvial discharge, a periodic sand or gravel spit (narrow with low-elevation) built by long-shore drift can occur. The dynamics of the spit can partially or fully close the river outlet

during the dry season when the wave-driven longshore sediment transport surpasses the ability of the river flow to remove sediment from the channel [18–20]. Therefore, a periodic evolution at a seasonal to pluriannual time scale of the outlet location and its open/close configuration can be observed. Such developments are not without consequence for the management of these environments. The degree of closure results in the increased vulnerability of coastal areas to the risk of flooding [5]. However, sand spits are also efficient natural coastal defenses, acting as protective barriers during marine storm events. These environments are commonly referred to in the literature as Intermittently Open/Closed Estuaries (IOCEs) [18], Intermittently Closed/Open Lakes and Lagoons (ICOLLs) [3,21], Temporarily Open/Closed Estuaries (TOCEs) [22,23], bar-built estuaries [24], or seasonally open inlets [19]. In the present study, we refer to these systems as Intermittently Open/Closed Estuaries (IOCEs).

Accurate prediction of the occurrence and morphological consequences of both fluvial floods and marine storms on coastal river outlets is of obvious importance for coastal flood risk assessment and management and erosion mitigation strategies. This knowledge is also crucial in the context of climate change and prediction of the increase in high energy events (floods, storms and surges) [25–27]. Fluvial floods and marine storms are common energetic events that drastically modify the morphology of the outlets and the sediment circulation pattern, in particular at IOCEs. Fluvial flood events provide a large amount of fresh water and sediments exported within the river jet and the surface plume in the nearshore zone. The coarser sediments are directly deposited on the delta close to the outlet, and the finest sediments are transported across the shoreface and deposited by flocculation processes [28,29]. An erosion of the river channel and the outlet also occurs by the action of jet scouring [30]. In contrast, during marine storm events, sediment transport is controlled by waves [31], which can transfer a large amount of sediment from the delta to the river channel when the outlet is no-barred [30]. In cases where a sand spit is present, fluvial flooding can lead to its breaching, and waves can erode or even breach it by overtopping and/or overwashing. This can induce river jet flushing [32] and/or large sand bypass amounts [33,34]. Breaching of the sand spit can be controlled by either fluvial flood and/or marine storms, but the impact of their concomitance remains poorly understood. Indeed, due to the presence of a temporal shift ranging from 1 and 13 days between the storm and flood peak, most of these events have been studied separately and not concomitantly, although they are linked [35,36]. This lack comes partly also from the difficulty of monitoring this type of environment by means of in situ measurements. While their survey on the basis of topographic observations has been more frequent in recent decades thanks to the development of instruments such as LiDAR, UAV, and video [14,37–39], the recording of hydrodynamic processes inside and outside the outlet remains a challenge and is relatively scarce, especially during high energy periods (fluvial and/or marine) [9,40]. This type of data is, however, fundamental to linking the observed morphological evolution, estimating transport rates by empirical formulas or feeding numerical models [41,42]. Consequently, the discrimination and influence of each control parameter are difficult to determine, and many more case studies from various and contrasting environments are needed for a better morphodynamic characterization of river outlets.

The aim of this study is to contribute to a better understanding of the impact of concomitant fluvial flooding and marine storm during an extreme meteorological event on a typical small Mediterranean river mouth (Têt River, SE France). To address this aim, monitoring using a combination of high-resolution hydrodynamic measurements and topographic and bathymetric surveys was performed before, during and after the high-energy storm "Gloria", which occurred in January 2020 along the SW Mediterranean coasts, and induced dramatic damages [43]. The overall study had three primary objectives: (i) evaluate the impact of hydrodynamic conditions on sand spit evolution and determine the mechanisms responsible for its destruction, (ii) estimate and quantify the transport pathways during spit breaching, and (iii) estimate the behavior of the sediment transport pattern in the nearshore.

## 2. Regional Setting

### 2.1. The Têt Fieldsite

The Têt River is a small mountainous river (about 100 km long) located in the SW of the Gulf of Lions (southern France), along the Mediterranean coast (Figure 1a). It drains a catchment basin extending over 1400 km² and is impacted by two dams, one of which is the dam reservoir of Vinca, located 55 km from the field site. The Têt River is subjected to long dry periods punctuated by flash flood events, typical of the Mediterranean climate regime. Annual precipitation is, on average, 750 mm year$^{-1}$ [44], and usually occurs during short but intense rainfall events. The mean water discharge is 11 m³ s$^{-1}$, but can exceed 1800 m³ s$^{-1}$ under high-energetic conditions [28], inducing an export of 90% of annual water and sediment flux to the coastal zone within a small number of days [44]. The site is a microtidal, wave-dominated environment [45]. The tidal range is very low (<0.30 m at mean spring tides). Nevertheless, large variations in sea water level can occur and can exceed 1 m near the shore [46]. Mean offshore significant wave heights ($H_S$) are generally low ($H_S$ mean = 0.67 m with $H_S$ < 0.3 m for 75% and $H_S$ < 1.5 m for 94% of the time, respectively), but can exceed more than 4 m during winter storms [40,45], and more than 7 m during the most energetic events. The prevailing wave direction from S-E leads to a residual northerly alongshore sediment drift estimated at 200,000 m³ year$^{-1}$ [47].

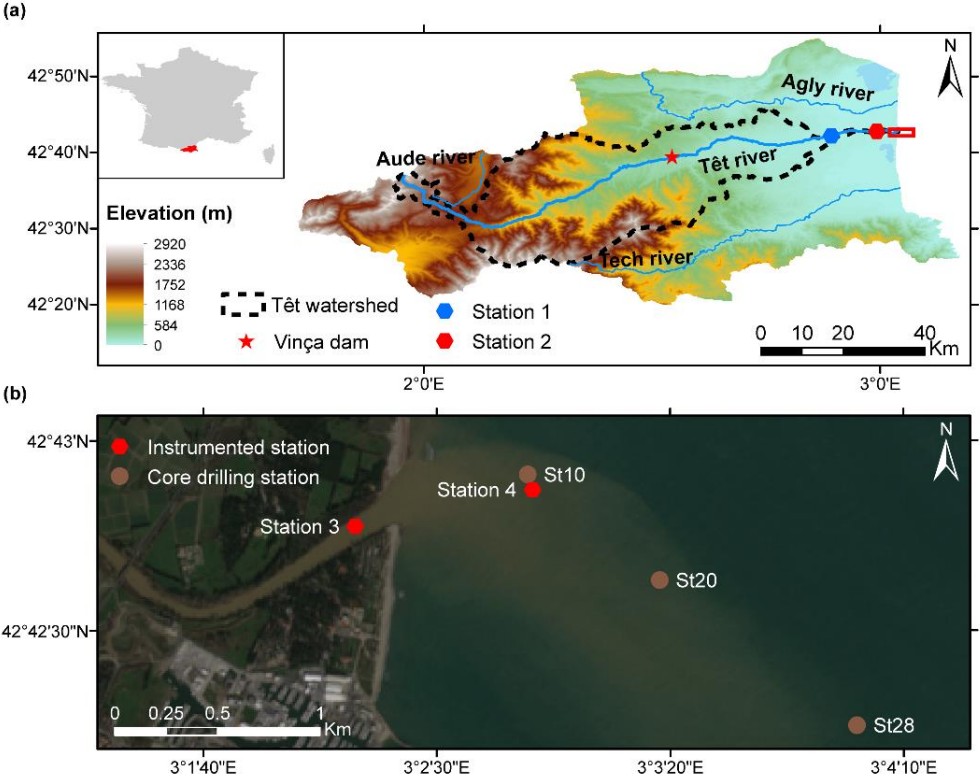

**Figure 1.** Location of the study area: (**a**) the position of hydrologic stations, the red frame corresponds to a zoom of (**b**) Sentinel 2 image measured on 28 January 2020, including the position of hydrodynamic stations and the core drilling location.

The Têt River mouth (entire area) is 150 m wide (cross-section), but is often shorter as a result of the dynamics of one or two sand spits reducing its opening (around 20 m) and modifying the location of the outlet (channel between the spits separating the river mouth from the sea). The fluvial side of the river mouth is very shallow, on average 1 m deep, and can reach 1.20 m in the northern part (Figure 1b). The outlet can quickly migrate alongshore by sand spit development under prevailing SE waves (northwards migration) or during offshore wind (Tramontane), generating short NE waves (southwards migration). The amplitude of the outlet migration can reach 1 km, causing the partial closure of the

river mouth [40]. This pattern matches the definition of Intermittently Open/Closed Estuaries (IOCESs) [18]. The sand spit is narrow (20 m) with low elevation (1 to 2 m above mean sea level), making it highly vulnerable to overwash processes during marine storms. The subaqueous delta is quite close to the outlet, with a very limited offshore extension (<100 m). The granulometry of the upper beach presents a $D_{50}$ of 0.67 mm and 0.3 mm on the nearshore [45]. The adjacent beach is classified as an intermediate barred beach with two nearshore crescentic bars [48]. The inner bar can be interrupted by rip channels and the outer bar is uninterrupted with oblique crescents [49].

### 2.2. Storm Gloria Event

Storm Gloria was a low-pressure atmospheric system that formed northward of the Azores Islands on 17 January 2020 and made landfall in the northwestern part of the Iberian Peninsula. In the subsequent days, this system moved towards the south-east until it reached the western Mediterranean Sea on 19 January, where it intensified, severely affecting the northern and eastern regions of the Iberian Peninsula, including the Balearic Islands and the Gulf of Lions. On 20 to 23 January, an intense anticyclone over the British Isles (1050 hPa, setting the historical record measured by the United Kingdom MetOffice since 1957 [43]) induced a strong barometric gradient between the British Isles and the south of Spain. It generated strong north-easterly winds, which caused a marine storm and intense rain, inducing severe flooding. Finally, the low-pressure system was absorbed by a larger low-pressure system located over the Alboran Sea, and lasted until 26 January [50,51]. Storm Gloria was characterized by a striking pattern of extreme waves (leading to record-breaking waves heights and periods in the western Mediterranean), storm surge (beating the record along Valencia's coastline), rain (return period: 20/30 years with a secular return for the month of January), and fluvial discharges that caused unprecedented erosion of both beaches and mouths, and several casualties, particularly in Spain [43,50,52,53].

## 3. Materials and Methods

As part of a regional monitoring program (DEM'EAUX) with the aim of studying the impact of meteorological events on the coastal area, hydrodynamic, topographic and bathymetric data were acquired and used to analyze the impact of an extreme storm and flood event (the Storm Gloria event) on the Têt River mouth.

### 3.1. Meteorological Data

Atmospheric data were analyzed from two meteorological stations of the French national meteorological service (Météo-France, https://donneespubliques.meteofrance.fr/, accessed on 26 November 2021) located at Perpignan airport (14 km west of the study site, hourly atmospheric pressure data) and Torreilles (7 km N-W of the study site, hourly wind speed and direction). Daily rainfall data were issued from SAFRAN, a mesoscale data analysis model on a regular 8 km grid [54]. Surface liquid and solid precipitation were added in order to obtain the total precipitation over the entire Têt River catchment.

### 3.2. Hydrodynamic and Sediment Flux Data

Four hydrodynamic stations (named 1 to 4) were implemented along the river (1, 2 and 3) and on the nearshore (4) to obtain water levels, water discharge, current velocities, wave data, and concentrations of suspended particulate matter.

### 3.2.1. Station 1 (Hydrodynamic)

Station 1 is located at Perpignan (13 km upstream from the outlet) and consists of a gauging station (code Y0474030). Hourly water levels and water discharge have been measured since the 1980s, and data are provided by the HYDRO database (http://www.hydro.eaufrance.fr/, accessed on 22 November 2021).

### 3.2.2. Station 2 (Sediment Flux)

Station 2 is located at Villelongue de la Salanque (4.5 km upstream from the outlet). Measurement devices include a pumping system and a refrigerated sampler Teledyne ISCO 6712FR allowing hourly surface water sampling. The suspended particulate matter (SPM) contained in the water samples was analyzed using a Malvern Mastersizer 3000 after five minutes of ultrasonification to determinate the primary particle size distribution (PSD). The estimation of PSD was performed using the Mie theory to describe suspended particles between 0.01 and 3500 μm. Each sample was measured five times, and data were then averaged.

To determine mass concentrations, water samples were filtered in triplicate on pre-weighed 0.7 μm GF/F Whatman filters. Then, the filters were rinsed with deionized water, dried at 50 °C, and re-weighed. The data were averaged to determine the absolute mass concentration of SPM and uncertainties in the measurements. The sand mass concentration was then calculated by applying the percentage of sands determined by the PSD on the absolute mass concentration of SPM.

A least squares regression method was used to estimate the relationship between the SPM and sand mass concentration in suspension (g L$^{-1}$) with the water discharge (m$^3$ s$^{-1}$) (Equations (1) and (2)).

$$[SPM] = 1.8946 \, (\pm 0.088) \times water \; discharge \left( r^2 = 0.90 \right) \tag{1}$$

$$[Sand \; mass \; concentration \; in \; suspension] = 0.2247 \, (\pm 0.018) \times water \; discharge \\ (r^2 = 0.76) \tag{2}$$

These relations were finally applied to calculate the total SPM and sand flux, in terms of both whole mass and volume (a density of 1.6 t m$^{-3}$ was used for sand [55]), occurring during the event in the river.

For the estimation of the bedload, Ref. [56] estimated that it represents 10% of the total load. Ref. [28] confirmed this value of 10% for the Têt River, and estimated the proportion of sand within the bedload to be 90% [57]. These percentages were applied to estimate the whole mass and volume of the bedload.

### 3.2.3. Stations 3 and 4 (Hydrodynamic and Sediment Flux)

Station 3 is located 200 m upstream of the outlet, and Station 4 at a depth of 10 m on the nearshore at a distance of 800 m from the shoreline in front of the outlet. Each station was equipped with an Optical Backscatter Sensor (Campbell Scientific OBS 3+) and an Acoustic Doppler Current Profiler (ADCP) (Figure 2). During the survey, Station 3 was destroyed by the high level of water discharge, accompanied by river flow debris, just before the fluvial peak. To fill in these missing data, the numerical X-Beach model (eXtreme Beach behavior [58]) was used to simulate the water fluvial level (for model validation, see [59]).

The current profiler of Station 3 (Figure 2a) is an ADCP Aquapro 1 MHz (©Nortek) fixed 30 cm from the bottom and oriented upwards, providing time series by averaging 60 s of measurements of pressure, temperature and vertical velocity profile (intensity and direction) every 10 min. Pressure is transformed into depth and then into water level, which was adjusted from the French National Geodesic referential (NGF) using in situ bathymetric data. The vertical velocity profile, with 30 cells of 10 cm each, was re-processed for beam-mapping using the Storm software (©Nortek).

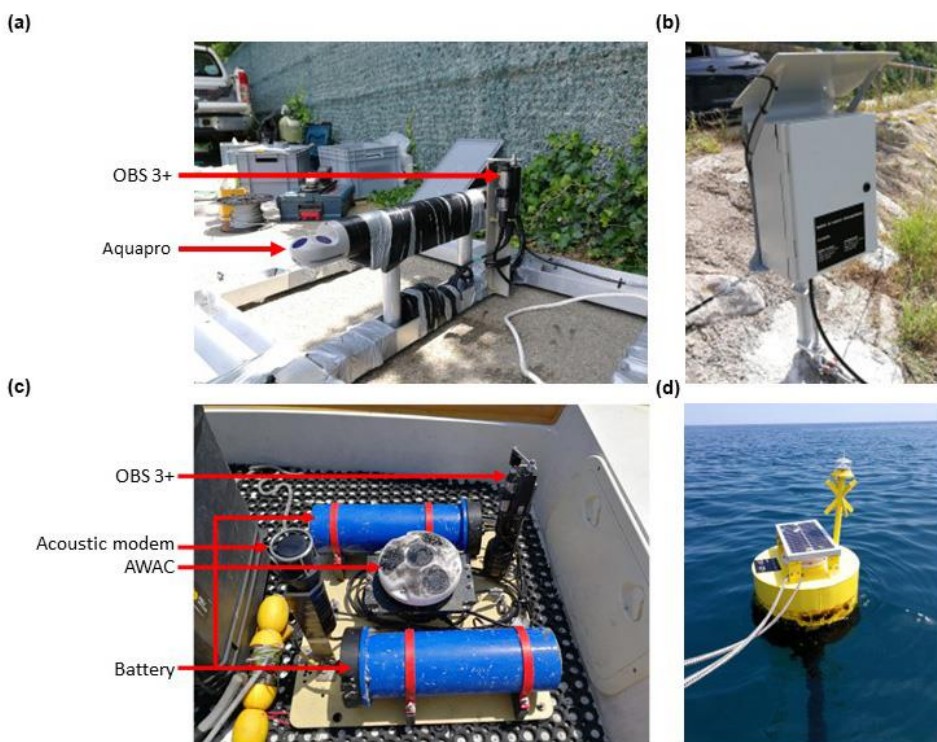

**Figure 2.** Instrumented stations on the Têt mouth: (**a**) Station 3 instrumented cage; (**b**) power supply and data transmission box; (**c**) Station 4 instrumented plate; (**d**) data transmission surface buoy.

The current profiler of Station 4 (Figure 2c) is an ADCP AWAC 1 MHz (©Nortek) fixed on the bottom and oriented upwards, providing time series averaging 60 s of measurements of pressure, temperature and vertical velocity profiles (intensity and direction) every 30 min. Water depth was derived from pressure using the Thermodynamic Equation Of Seawater equations (TEOS-10) [60]. The water level was then adjusted to the tide gauge measurements of Port-Vendres (https://data.shom.fr/, accessed on 24 November 2021) to obtain a time series based on the NGF referential. The vertical velocity profile, consisting of 10 cells of 1 m, was re-processed for beam-mapping using the Storm software (©Nortek). The AWAC also provides wave data ($H_S$ and direction) every 30 min (510-s burst average). A calibration of the optical signal of the OBS was performed in the laboratory using in situ samples collected at a depth of 10 m close to Station 4, in order to estimate SPM concentrations. Samples were used to establish a concentration range from 0 to 0.1, 0.1 to 1, 1 to 10 and 10 to 15 g L$^{-1}$ in intervals of 0.05, 0.1, 1 and 5 g L$^{-1}$, respectively. Each concentration was agitated and exposed at the OBS connected to a multimeter. The second-order polynomial regression method was used to estimate the relationship between the mass concentration of SPM (g L$^{-1}$) and the voltage of OBS (Equation (3)).

$$[Sea\ SPM\ concentration] = 516.45x^2 + 3096.6x + 22.309\ \left(r^2 = 0.99\right) \tag{3}$$

where $x$ is the voltage of OBS (V).

### 3.3. Wave Set-Up and Run-Up

A time series of the highest elevation of wave set-up ($S_{high}$) and run-up ($R_{high}$) was calculated. They were estimated as the sum of the still water level of the sea, $\eta$, including tide and surge, and the set-up and run-up calculated using the formulas presented in [61].

$$S_{high} = \eta + 0.35\beta_f(H_0L_0)^{1/2} \tag{4}$$

$$R_{high} = \eta + 1.1 \left( 0.35\beta_f (H_0 L_0)^{1/2} + \frac{\left[ H_0 L_0 \left( 0.563\beta_f{}^2 + 0.004 \right) \right]^{1/2}}{2} \right) \quad (5)$$

where $\beta_f$ is beachface slope (0.05), $H_0$ is significant deep-water wave height and $L_0$ is the deep-water wave length. Offshore wave conditions were recorded by the CANDHIS network at the Banyuls buoy (CEREMA/DREAL Occitanie/Observatoire Océanologique de Banyuls) moored to the south of the study area at a depth of 50 m. The sea water level was recorded by the REFMAR network at the tide gauge of Port-Vendres (https://data.shom.fr/, accessed on 24 November 2021).

### 3.4. Collection and Analysis of Topographic/Bathymetric Data

The LiDAR dataset is composed of two surveys on 04 November 2019 and 28 January 2020, conducted by a LiDAR YellowScan VX20 mounted on a DJI M600 UAV. N-S and E-W amplitudes for the LiDAR flight were 850 and 250 m, respectively (0.215 km²) with a final resolution of 10 cm. The Z accuracy was ±0.025 m. Additional topographic data were monitored using a kinematic DGPS Trimble R6, with an accuracy of ±0.05 m, on 15 November 2019 and 30 January 2020.

Two bathymetric surveys were carried out on the river side of the mouth on 15 November 2019 and 30 January 2020, and two others were conducted in the nearshore area on 08 April 2019 and 13 February 2020. A single-beam echosounder Tritech PA500 coupled to a RTK-GNSS antenna Trimble R6 or R8 was used, working in VRS mode, through Hypack software (Z accuracy of ±0.1 m). Cross-shore and longshore survey lines were measured from 300 m upstream to the river mouth to a depth of −14 m in the nearshore zone. Outliers and spikes were manually removed from the output files and smoothing was performed (7 times the wave period) using Hypack® software.

To compute the volume changes, Digital Elevation Models (DEMs) were generated using a GIS software package (ESRI ArcGIS®) with the natural neighbor interpolation method.

### 3.5. Sediment Cores

Three sediment cores (namely, St10, St20 and St28 in Figure 1b) were collected in triplicate by SCUBA divers using transparent Perspex tubes (20 cm length, 4 cm diameter) at water depths of 10, 20 and 28 m on 12 February 2020. Each core was described and sectioned into 1-cm-thick slices, except for the first cm, which was sectioned into two 0.5 cm layers. Grain-size analyses were performed on sonicated samples for 5 min with MilliQ-filtered water using a Malvern Mastersizer 3000 particle size analyzer. The radioisotope ⁷Be and ²¹⁰Pb activities were determined at the LAFARA gamma spectrometry laboratory (Université Toulouse III Paul Sabatier, https://lafara.obs-mip.fr/, acessed on 13 January 2022). The samples were analyzed on gamma spectrometers equipped with electric cooling systems. The spectrometers were placed underground under 85 m of rock, thus protecting them from cosmic (see [62] for detailed methodology).

## 4. Results

### 4.1. Local Characterization of Storm Gloria

The Gloria event impacted the Têt River, and its area from 18 to 26 January 2020 and was characterized by intense precipitation. Most rainfall was concentrated in the SE part of the Têt catchment area, upstream of the Vinça dam (cumulative rainfall during the event was >300 mm, with a maximum of 560 mm), whereas downstream of the dam, rainfall was less significant (between 130 and 250 mm) (Figure 3).

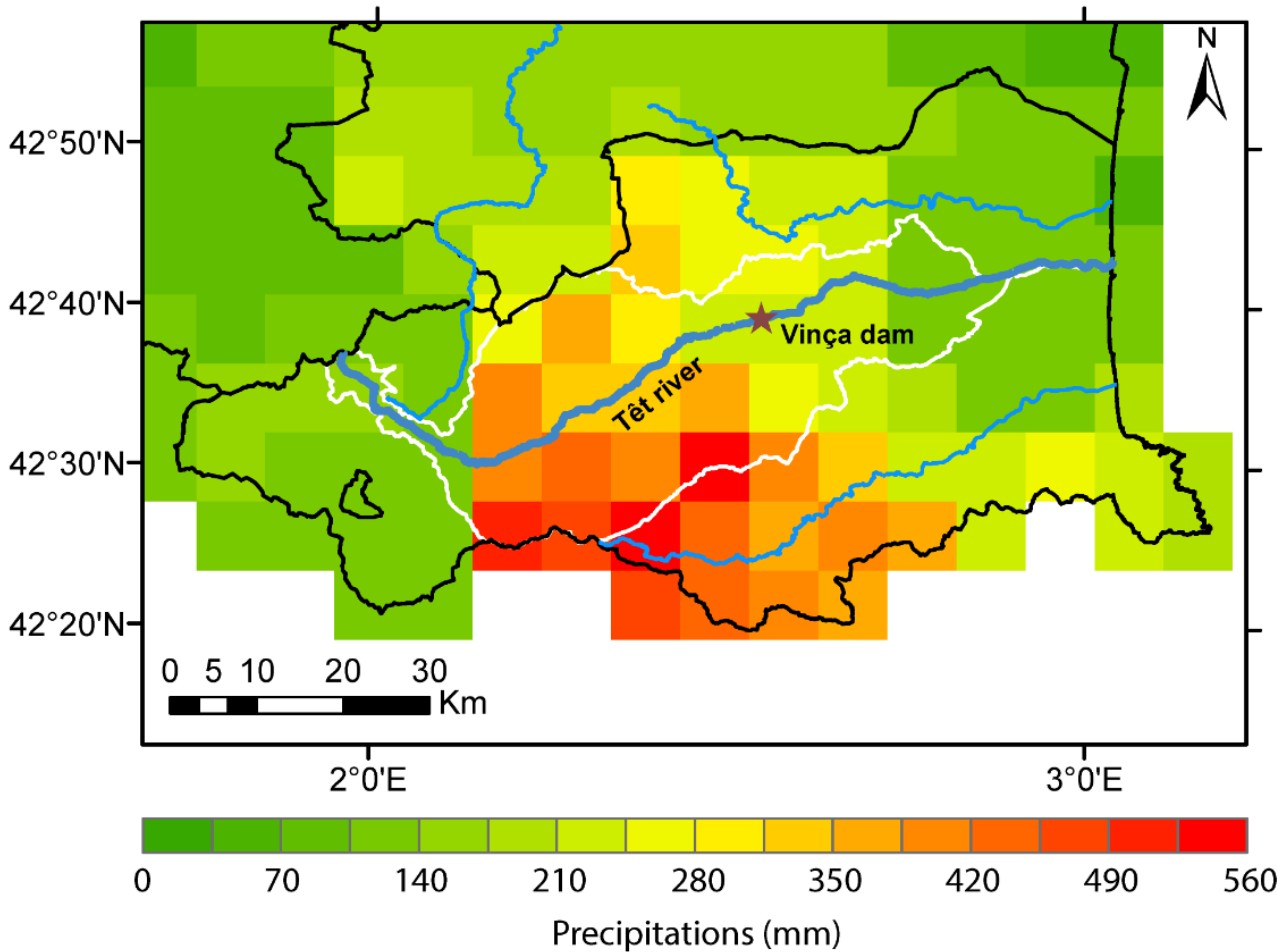

**Figure 3.** Accumulated precipitation from 18 to 26 January 2020 on a grid of 8 km resolution (SAFRAN). The limits of the French departments are indicated by black polygons, the Têt catchment area by white polygon, and the rivers by blue lines. The position of the Vinça dam is indicated by the brown star.

The meteorological conditions first indicated a decrease in atmospheric pressure between 18 January at 09:00 UTC and 19 January (1031 to 1022 hPa) (Figure 4a), accompanied by fairly weak easterly winds (2.3 to 5.4 m s$^{-1}$) (Figure 4b). Until 21 January, a rapid increase in atmospheric pressure (to 1036 hPa) was observed without significant change in wind characteristics. From 21 to 24 January, a second decrease in atmospheric pressure occurred, with moderate winds (up to 12 m s$^{-1}$) and a shift from an easterly to an S-E direction. After 24 January, the atmospheric pressure stabilized around 1022 hPa, and the winds became weak. Precipitation, mostly concentrated on 21 January, led to rapid increase in the Têt River discharge from 5 to 1000 m$^3$ s$^{-1}$ (22 January at 16:00) (Figure 4c). After a decrease in the river discharge over a small number of hours, a second peak reaching 1280 m$^3$ s$^{-1}$ was observed on 25 January at 12:00 H (according to the HYDRO database, the return period of this flood event was estimated to be more than 50 years). Finally, the water discharge decreased progressively until to 26 January. The H$_S$ was below 1.5 m (East to N-E direction) until 20 January at 08:00 (Figure 4d). From there, a rapid increase was observed, reaching 6.3 m (according to the CANDHIS database, the return period was estimated to 15 years for the zone, [63]) 21 January at 12:00. The decrease in H$_S$ started on 22 January at 07:00 and lasted until 26 January, with waves coming from the southeast.

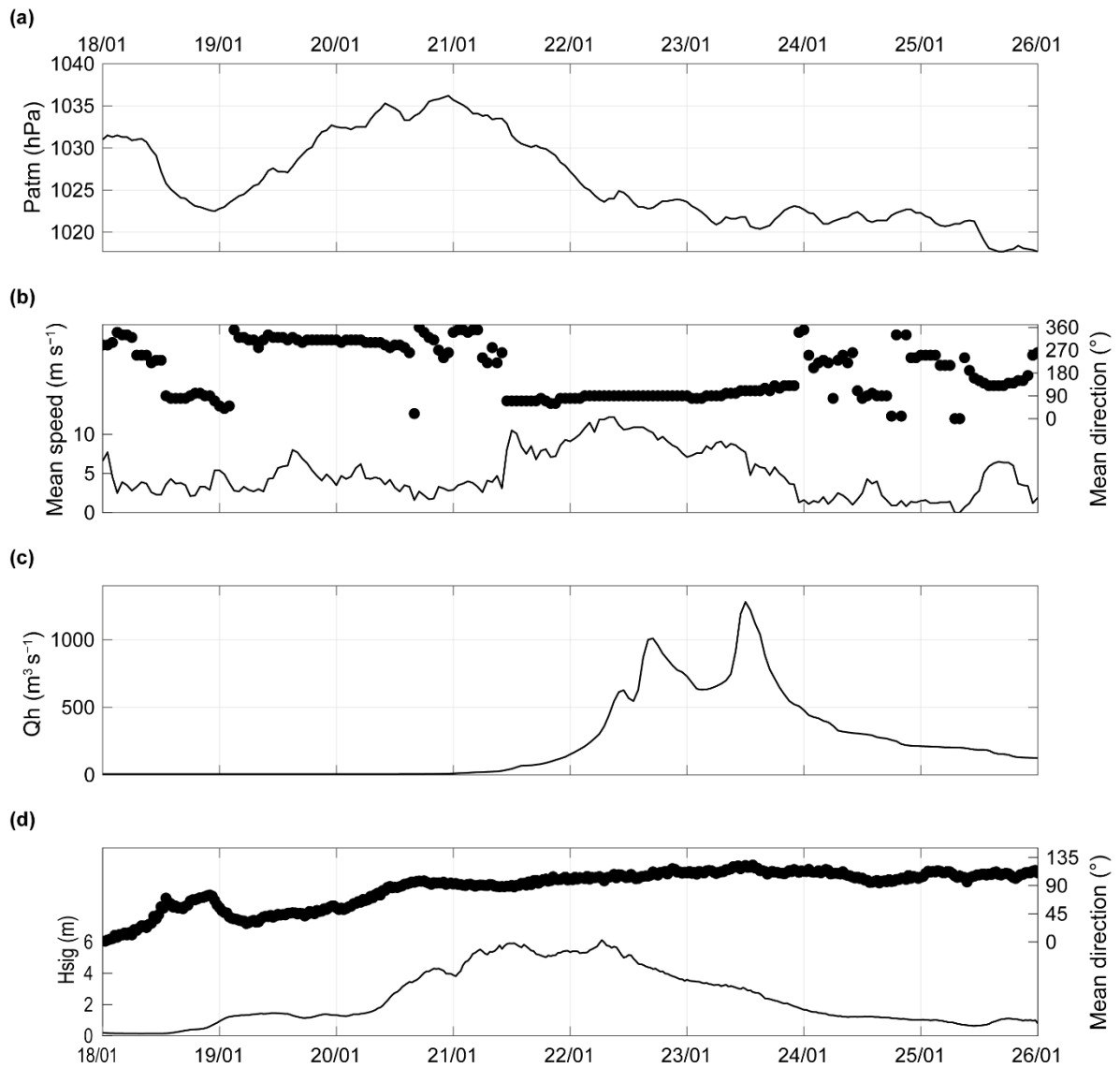

**Figure 4.** Time series from 18 to 26 January 2020 of: (**a**) local atmospheric pressure (Perpignan airport), (**b**) hourly mean wind speed and direction represented by black line and dots, respectively (Torreilles station), (**c**) Têt hourly discharge (Station 1), and (**d**) significant wave height and direction represented by black line and dots, respectively (Station 4). By convention, wind direction indicates its origin, and the $H_S$ the direction of its flow.

### 4.2. River Mouth Hydrodynamics

Analysis of $H_S$ data made it possible to highlight three periods during Storm Gloria: the waxing storm, the storm peak, and the waning storm (Figure 5).

Waxing storm: During this period ($H_S$: 0.16 to 5.5 m), sea water level, set-up and run-up max showed increases of 0.4, 0.7 and 1.9 m, respectively, with a slight tidal signature (Figure 5a,b). The fluvial level of the Têt River was characterized by different behaviors between Stations 1 and 3. The first station presented a relatively stable fluvial level until 21 January (0.3 m), whereas Station 3 showed an increase by 1 m from 19 to 21 January at 06:00 (Figure 5c). River currents in this last station were weak (0.1 m s$^{-1}$) and oriented to the north (Figure 5d,e). The concentration of SPM measured at Station 4 at a depth of 10 m first showed an increase from 19 January at 21:00, and remained moderate (around 6 g L$^{-1}$) (Figure 5f).

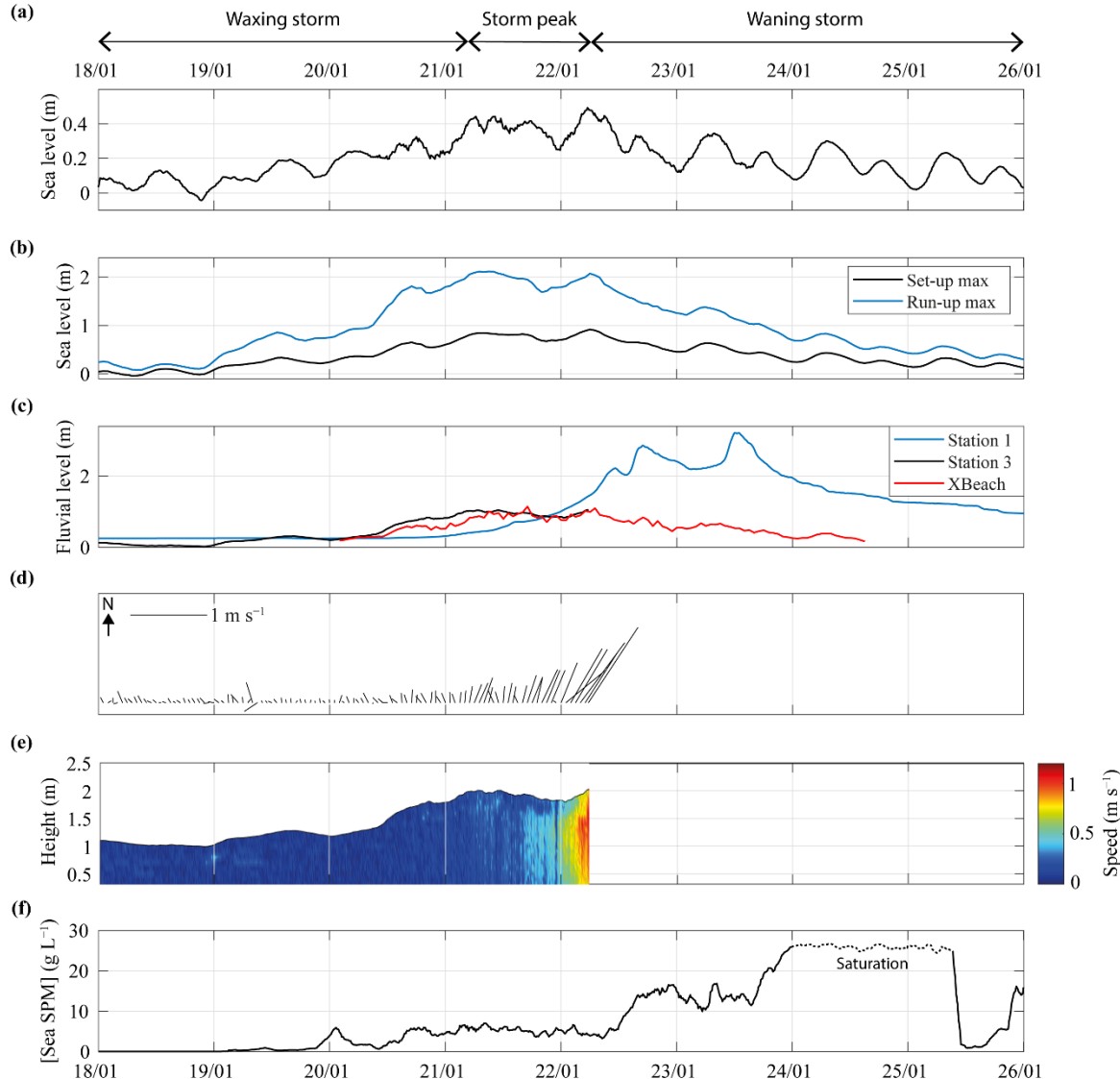

**Figure 5.** Time series from January 18 to 26, 2020 of: (**a**) sea water level (Station 4), (**b**) calculated set-up and run-up max, (**c**) fluvial water level (each stations are indicated in legend), (**d**) mean current direction and velocity in the inland mouth (Station 3), (**e**) velocity in the water column (Station 3) and (**f**) concentration of SPM measured offshore at Station 4 at a depth of 10 m depth (Station 4). By convention, current sticks represent the direction towards which the current is flowing.

Storm peak: The peak of the storm was characterized by an $H_S$ of 5.5 m for 25 h. This induced high values of sea water level, and set-up and run-up max exhibited maximum values of 0.5, 0.92 and 2.1 m (Figure 5a,b). Fluvial level started to increase at Station 1 and remained stable up to 1 m at Station 3 until its destruction (Figure 5c). River current velocity at Station 3 showed a rapid and brutal increase (0.3 to 1 m s$^{-1}$ in 6 h), with an orientation towards the North-East (i.e., the main river channel direction) (Figure 5d,e). The concentration of SPM at Station 4 remained moderate, at around 6 g L$^{-1}$ (Figure 5f).

Waning storm: During this period ($H_S$: 6.3 to 0.6 m), sea water level, set-up and run-up max decreased progressively to 0.1, 0.1 and 0.3 m, respectively, in relation to decreasing wave conditions (Figure 5a,b). An increase in fluvial level at the station continued until 2.9 m on 22 January at 16:00, before falling to 3.2 m on 23 January at 12:00 (peak of the fluvial flood). After that, the Têt's level decreased progressively until to 26 January. Model outputs at Station 3 showed that only during this a period did the fluvial level decrease (Figure 5c). A first rapid increase in SPM was recorded at Station 4 (up to 15 g L$^{-1}$); then, a second

occurred on 23 January at 15:00 (up to 26 g $L^{-1}$), saturating the sensor. On 25 January, the concentration rapidly decreased (down to 0.9 g $L^{-1}$ in 3 h).

　　Figure 6a shows the important morphological evolutions of the Têt mouth with the northern spit breaching (Figure 6a) and the complete destruction of the mouth 2H30 before the peak of the fluvial flood (Figure 6b).

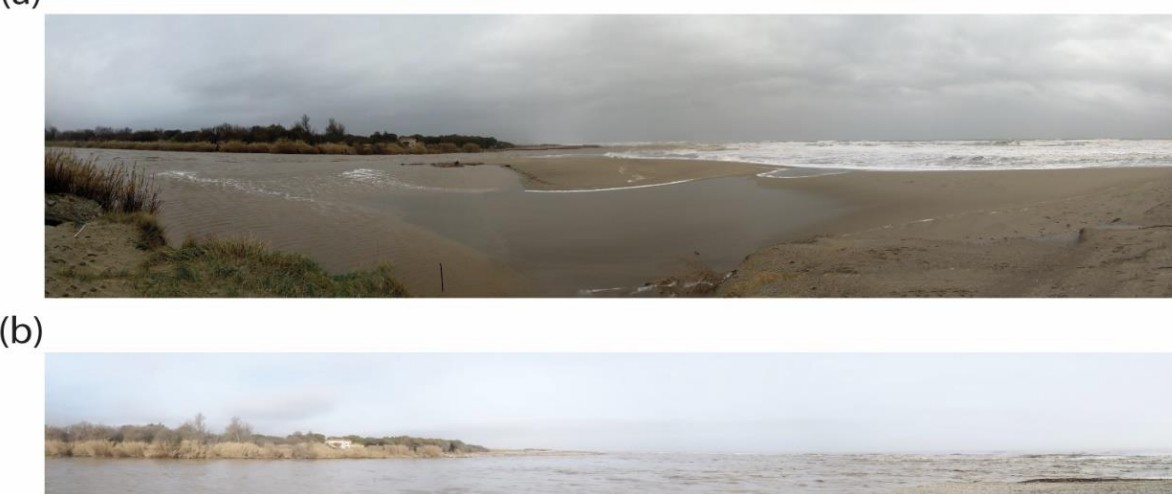

**Figure 6.** Morphology of the Têt mouth at (**a**) 21 January 2020 at 12:00 and (**b**) 24 January 2020 at 11:30.

*4.3. Morphological Changes*

　　Before the Gloria event, the Têt outlet was very narrow (20 m), and was partially closed by two sand spits on the southern updrift side (90 m long) and on the northern downdrift side (50 m long) (Figure 7a). Their maximum elevation was low, at around 1.2 m above mean sea level. The fluvial part of the mouth was characterized by a deep channel close to the northern bank of the river (up to 2.8 m water depth), with a very large shallow platform (around 1 m water depth) in the central and southern part. The nearshore zone was characterized by a small subaqueous delta in front of the main axis of the river, and a well-developed crescentic nearshore bar in front of the adjacent beach with the crest at approximately 5 m water depth. After the Gloria event, the total destruction of the spits occurred, and the outlet was 250 m wide and 1.5 m deep (Figures 7b and 8a). Sediment loss in this area is estimated to be around $-37,000 \pm 2800$ m$^3$ (zone 2). The water depth of the channel in the fluvial part increased due to an erosion of around 0.7 m ($8300 \pm 1900$ m$^3$, zone 3) (Figure 7a), while the rest of this part of the mouth did not change significantly. Important changes in the nearshore area were observed, with a large accretion zone with a thickness of 1.6 m, reaching a maximum of 2.7 m, just northwards of the outlet ($90,000 \pm 10,000$ m$^3$, zone 1) (Figures 7c and 8b,c). Its geometry was not uniform, with a steep slope on its north side and a gentler slope on its south side (Figure 8c). A second accretion zone was observed in shallow water along the updrift side of the mouth ($7200 \pm 900$ m$^3$, zone 4). During the event, the crescentic bar moved offshore by approximately 90 m.

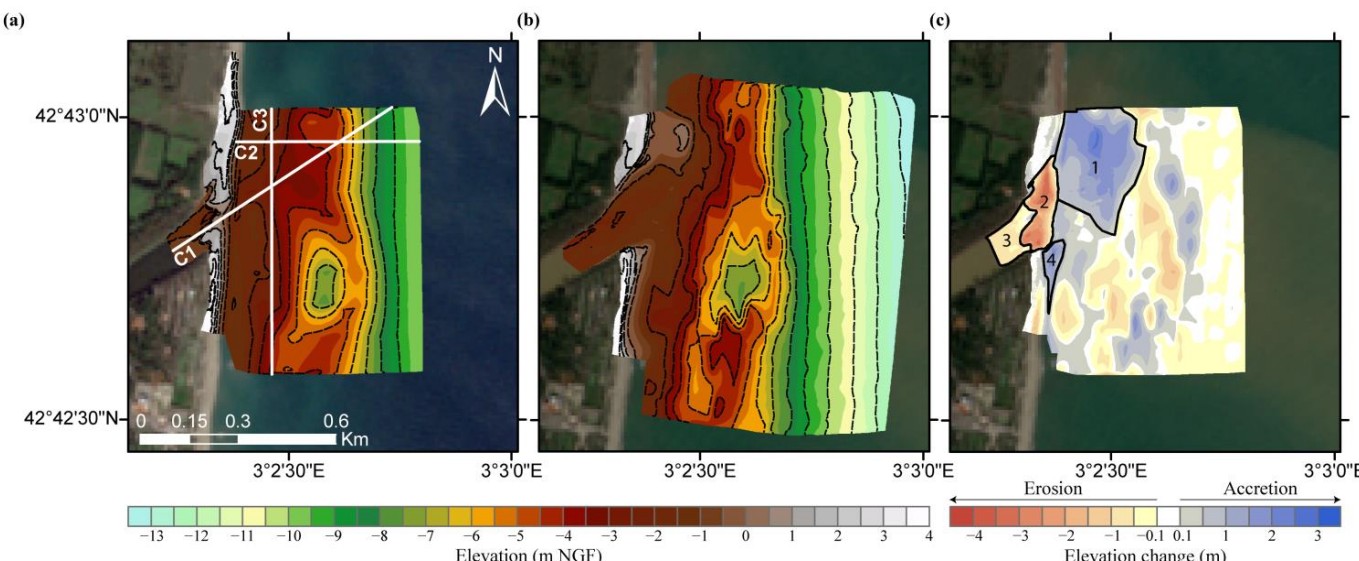

**Figure 7.** (**a**) The digital elevation model before the Gloria event, (**b**) the digital elevation model after the Gloria event, and (**c**) the differential digital elevation model acquired from the comparison of the two previous ones. Dashed and black lines correspond to topographic/bathymetric isocontours and the volume calculation areas numbered from 1 to 4, respectively. White lines correspond to the profile evolutions shown in Figure 8.

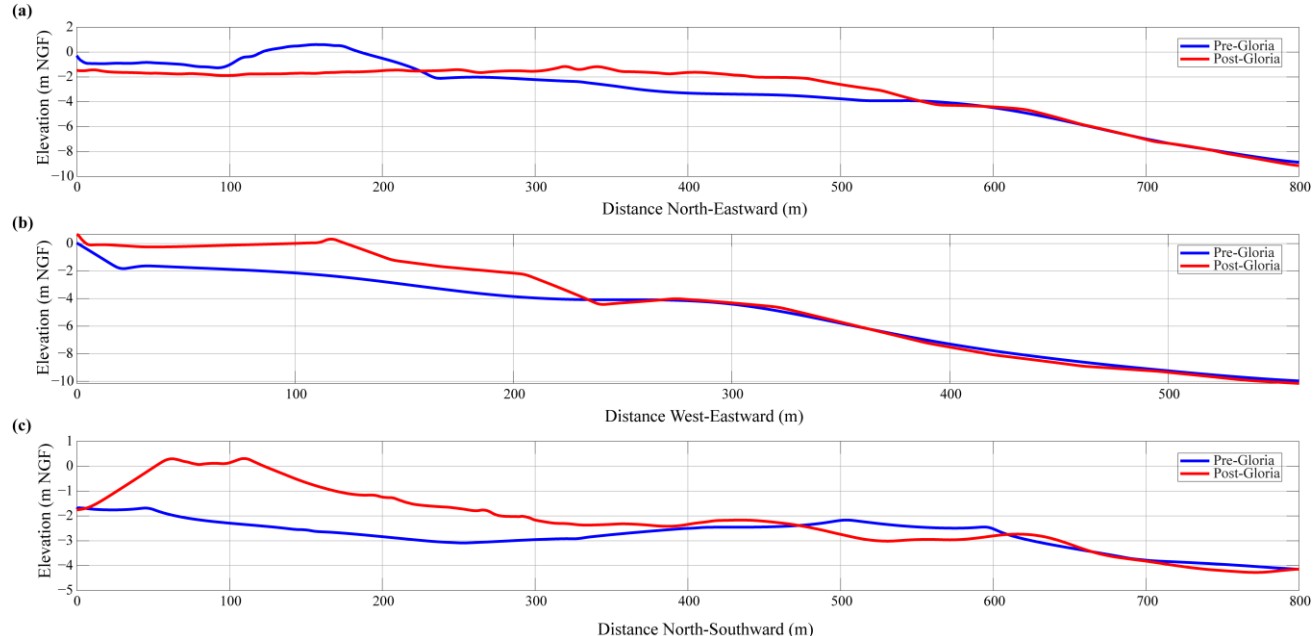

**Figure 8.** (**a**) Longitudinal profile evolution (C1), (**b**) cross-shore profile evolution (C2), and (**c**) long-shore profile evolution (C3).

### 4.4. Coastal Sedimentation Characteristics

Facies description and grain size analysis were performed on the three sediment cores at depths of 10, 20 and 28 m, in front of the Têt River (St10, St20 and St28 on Figure 1b. Analysis revealed four distinct sedimentary layers (Figure 9). The first layer was characterized by fluffy material, mostly composed of fine sediment (from 12 to 45% of >63 μm) with the fine proportion increasing with distance to the river mouth. Its thickness was 3 cm for St10, 2 cm for St20, and 0.5 cm for St 28. This layer was interpreted as being the Gloria flood deposit for St10 and St20. The second layer was only present in St10 (at depths from 3 to 7 cm). It was composed of a mixture of flood deposit (sand and silts (40%

of >63 µm)) and coarser sediments. The third layer was composed of sandy silts, with the proportion of sediment >63 µm increasing rapidly within the sediment depth for St10. Sediment became finer and the proportion of sediment >63 µm decreased progressively with increasing distance from the river mouth. This layer was present from 7 cm for St10, from 2 cm for St20, and from 0.5 to 4 cm for St28. The last layer was composed of mud (between 20 and 30% of >63 µm), and was only present in St28 below 4 cm. Reworked layers were estimated on the basis of the presence of excess $^{210}$Pb values in the sediment for St10, St20 and St28 at depths of 7, 2 and 0.5 cm, respectively.

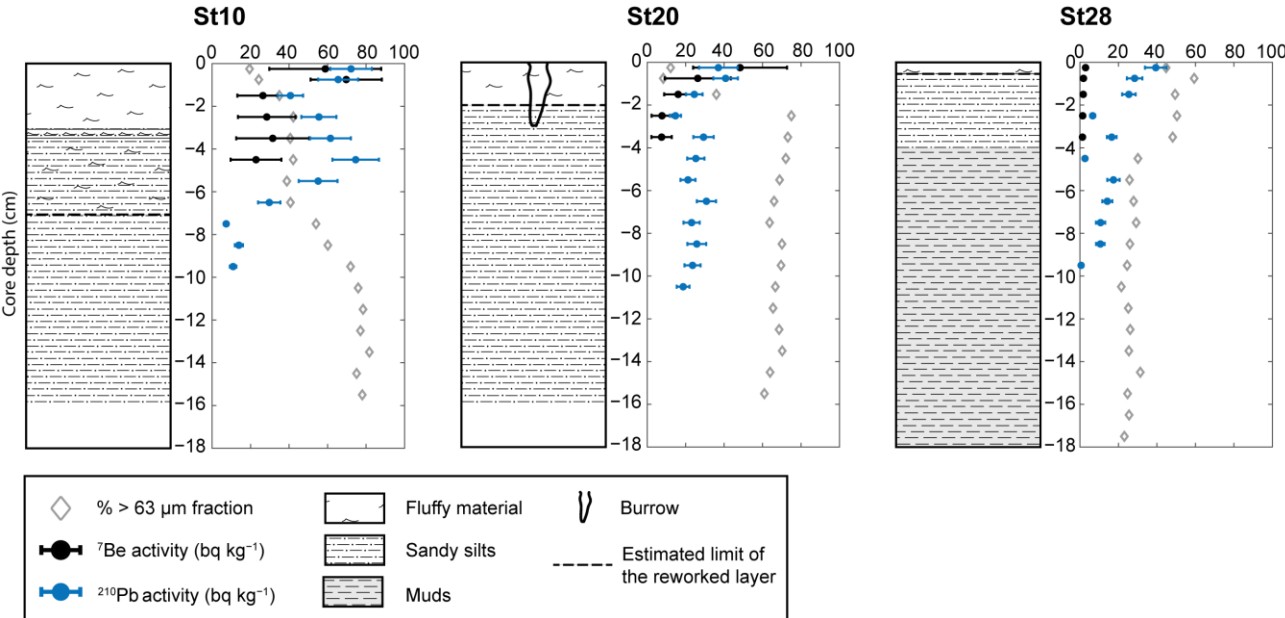

**Figure 9.** Sedimentary logs of sediment cores at 10, 20 and 28 m depth from 12 January 2020 showing down-core profiles of the fraction of particles >63 µm, the Be$^7$ and the $^{210}$Pb activities.

For the St10 core, the radioisotope $^7$Be (T$_{1/2}$ = 53.2 days) presented an activity of 70 Bq kg$^{-1}$, which decreased rapidly with increasing depth (up to 25 Bq kg$^{-1}$). Its activity was much lower in the St20 core, but with the same trend in depth (up to 7.5 Bq kg$^{-1}$). Little $^7$Be activity was found in the deepest core (St28) (maximum 2.9 Bq kg$^{-1}$).

The activity of $^{210}$Pb (T1/2 = 22.3 years) was more variable in the St10 core, which first exhibited a decrease, with values between 72 to 25 Bq kg$^{-1}$, followed by a slight increase before falling gradually with increasing depth (up to 7 Bq kg$^{-1}$). Its activity was much lower in the St20 core, but it followed the same trend with increasing depth (up to 20 Bq kg$^{-1}$). The activity of $^{210}$Pb in the St28 core showed a regular decrease with depth from 40 to 0.6 Bq kg$^{-1}$.

*4.5. Sediment Fluxes*

Table 1 summarizes the sediment fluxes with respect to the fluvial input estimated during the Gloria event. The total SPM reaching the sea during this flood was estimated to be 198,000 ± 28,000 t. The sand fraction was estimated to be 23,500 ± 600 t (14,700 ± 400 m$^3$), representing 12% of the total suspended load during this flood. The estimation of the bedload was to 22,000 ± 3100 t for an estimated sand volume of 19,800 ± 2800 t (12,400 ± 1800 m$^3$).

**Table 1.** Summary of the estimated sediment fluxes during the Gloria event.

| Type of Transport | Total (t) | Sand (t) | Sand Volume (m³) |
|---|---|---|---|
| SPM | 198,000 ± 28,000 | 23,500 ± 600 | 14,700 ± 400 |
| Bedload | 22,000 ± 3 100 | 19,800 ± 2800 | 12,400 ± 1800 |

## 5. Discussion

### 5.1. Conceptual Model of Spit Breaching during Fluvial/Marine Event

The monitoring of the Têt River mouth described in this study makes it possible to propose a conceptual model of its morphological adjustment in response to the concomitance of fluvial/marine processes during the Gloria event (Figure 10). This model is intended to complement the models proposed by [64–67], which focused mainly on the response of a sand/gravel barrier to marine process by adding the impact of a concomitant event. This model is divided into five stages. Stage 1 displays the initial morphology and hydrodynamic situation before the event. The fluvial and marine level are similar, allowing a natural outflow of the river even if the outlet width is reduced by the presence of the spits. During Stage 2, despite an increase in atmospheric pressure and offshore winds, an increase in sea level and significant wave heights is observed, which is associated with the swell conditions generated between France and Corsica Island. Due to a low elevation (1 to 2 m above mean sea level) and narrow shape (a few tens of meters width) of the spits, overtopping occurred on their lowest parts. This caused an increase in the water level in the fluvial part of the mouth, reinforced by the difficulty of outflow on the outlet due to breaking waves. During Stage 3, overtopping was more frequent, inducing erosion of the crest of the spits and an early phase of inundation. These overwash events induced a landward sediment transport (overwash deposits). At the same time, the seepage of marine and river water through spits and the saturation of the table probably contributed to its weakness. Stage 4 corresponds to the peak of the storm, marked by strong marine winds (E-SE). Coastal sea level, $H_S$, and overtopping reach their maximum. The precipitation that took place earlier on the watershed at this point initiated flooding of the river, helping to further increase the water fluvial level, leading to the breaching of spit. The magnitude and extent of this breaching cannot be estimated (except that it started in the north), due to the strong offshore velocities measured in the river mouth, attesting to the flushing flow from the river mouth to the nearshore area. Finally, Stage 5 is characterized by the decrease in marine processes and the peak of the flood, completing the total destruction of the spits. Despite high fluvial flows, incident waves (S-E) induced the deposition of the major part of exported sediment in the northward part of the mouth in shallow water, except for the finest sediment fraction, which was transported further on the nearshore. Marine processes contributed to the weakening of the spits and the initiation of their breaching, with the exceptional impact of the subsequent flooding helping to amplify the breaching, leading to the complete destruction of the mouth. This phenomenon demonstrates the complexity of such concurrent storm and flood events.

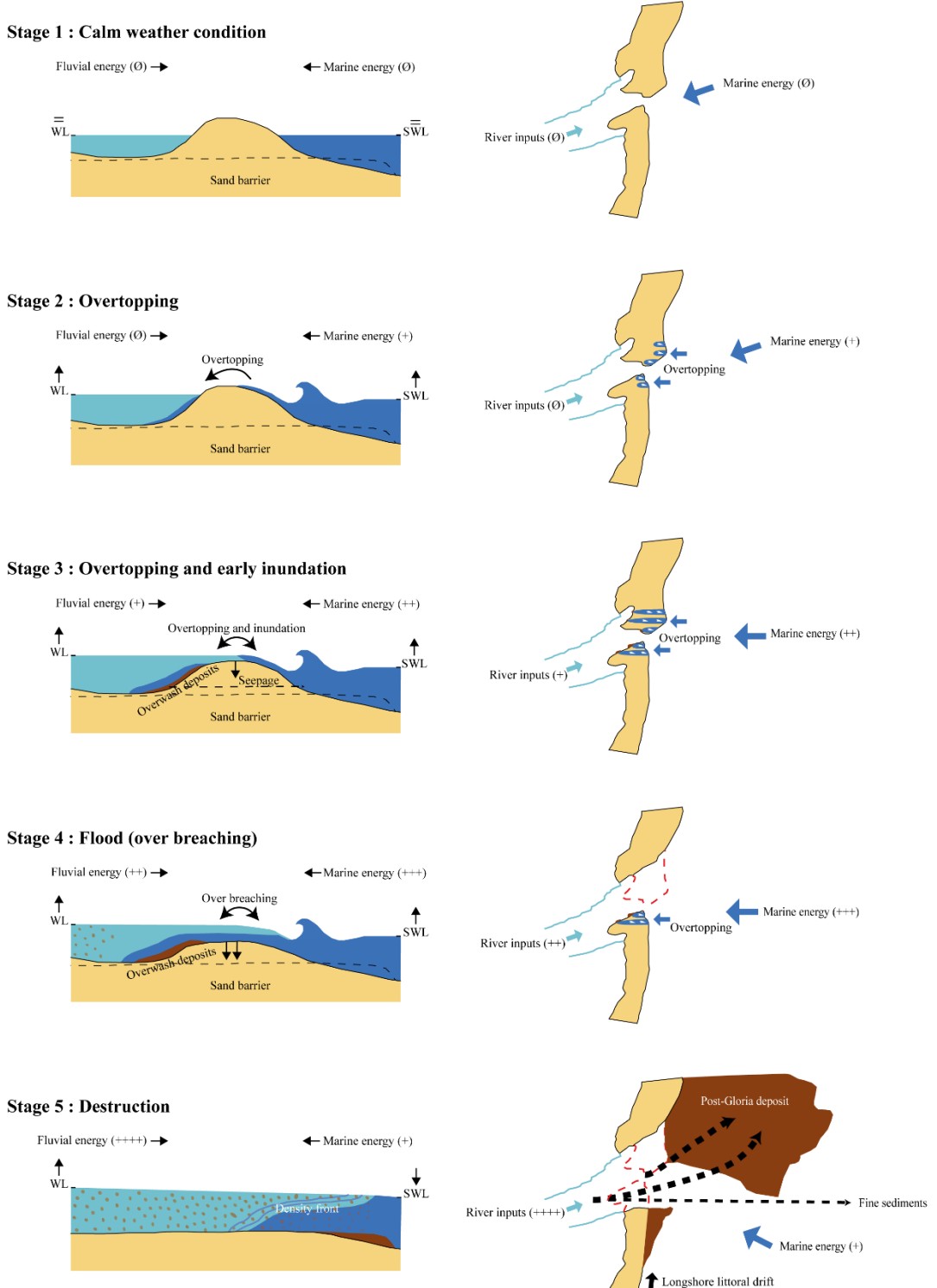

**Figure 10.** Conceptual model of evolution of the Têt River mouth during the Gloria event.

### 5.2. Role of Hydrodynamic Condition in Spit Breaching

Coastal flood risk assessments and morphological changes in river mouth areas are generally based on approaches in which fluvial floods [8,68] and storm surges [69,70] occur separately. However, they may additionally stem from a common meteorological cause, making these events concomitant. The small temporal shift between these two events (as in the case of the Gloria event) varies between 1 and 13 days depending on the

catchment area configuration (i.e., the presence of dams or other anthropogenic influences) or meteorological characteristics [35,36]. This is the case in the Gulf of Lions, where "wet storms" are typical, and are characterized by back-to-back storm and flood events [71]. Recently, these concomitant events have mainly been studied in terms of coastal flooding under the name of "Compound flooding" [36,72]. However, morphological response is less documented, while their adjustment is necessarily different and more complex. In this study, the small temporal shift between fluvial and marine processes enabled better discrimination of the influence of each process on the morphodynamics of the outlet.

The impact of marine processes on spit breaching is well known, whereby storm events lead the vulnerability to wave run-up and to overtopping over the coastal barrier [73,74]. Overtopping, the transport of large amounts of sediment landward in the form of overwash deposits [75], therefore contributes to reducing the height of the spit, thus weakening it [22,70]. Similarly, the transfer of water by overtopping and the perturbation of the outflow by the storm can participate in the establishment of a significant level shift between the two sides of the spit, which, by pressure and seepage, can lead to the initiation of a breach [32,76,77]. In our study, we observed the same processes at the beginning of the event, which led to the weakening of the sand spits and the initiation of their breaching. These marine processes were then limited by the oncoming flood, which reduced the wave action. The impact of flooding on spit morphodynamics is also well known, and can lead to the erosion of the river channel and the outlet [30]. When river discharge is sufficient, breach of the spit can be observed [9,78], resulting a rapid drop in the water level [79]. During extreme events, total destruction can occur, as in the Santa Clara River (California) [80] or during the Gloria event described in our study.

Our results show that storm and flood events may elicit opposite responses, particularly in terms of sediment transport, but both participate in spit breaching. When these events occur at the same time, great morphological evolutions can be observed, even in cases where the consideration of either driver alone would not be particularly severe [35]. This is the consequence of the concomitance of two events within a short space of time. Nevertheless, the main mechanisms explaining breaching remained unclear. The first works [40] on the Têt River mouth suggested that the river jet itself (for flood <241 m$^3$ s$^{-1}$) made a very small contribution to the opening of the river mouth. In this study, we confirmed the role of marine processes in the initiation of breaching. However, due to its exceptional intensity, the impact of the flood on the destruction of the outlet is clear. It flushed out the remaining spits and enlarged the channel at its maximum.

### 5.3. Sediment Transfers

The Gloria event led to the complete destruction of the Têt River mouth, with a large seaward sand transport to the nearshore. We observed the creation of a large subaqueous delta (zone 1 on Figure 7) that was shifted slightly northwards from the main axis of the channel. The same behavior was observed by [28] in a previous event, and could be the result of several mechanisms. During the destruction of the spit, the river jet becomes stronger than the longshore currents, and offshore circulation is jet-dominated [81], which is attested by the small accretion area in the south. In this case, a subaqueous delta is deposited in the direction of the river jet, following the axis of the channel (N-E). As shown in [82], the presence of bars, acting as bathymetric obstacles in front of the river jet, are able to modify local flow patterns to the point of suppressing the jet-like flow structure. Ref. [81] argued that these bars cause the spread of lateral momentum and the inhibition of vertical spreading. Therefore, the presence of bars disturbs the river jet, limiting its offshore extension and subjecting it to longshore current.

The sediment volume of this large subaqueous delta (zone 1) is estimated at 90,000 ± 10,000 m$^3$. Several hypotheses can explain its source. First, it can be determined that the erosion of the first 100 m of the inner mouth, as well as the destruction of the sandy spits, participated about 50% (45,300 ± 4700 m$^3$). In addition, it is estimated that the river inputs contributed to 30% by means of SPM and bedload transport. The estimated

input of SPM at Station 2 (198,000 $\pm$ 28,000 t over 8 days) corresponds to 4.4 times the total mean annual suspended load of the Têt River (45,000 $\pm$ 35,000 t year$^{-1}$ [83]). The sandy fraction represents about 12% of this amount (23,500 $\pm$ 600 t), which is a lower proportion than that observed on the Eel River, a larger mountainous river on the Californian margin ($\sim$24%, [84], or during the flood of 2004 on the Têt ($\sim$25%, [28]). This can be explained by the fact that the maximum rainfall was located upstream of the Vinça dam, unlike the flood of 2004, which occurred mainly in the lower part of the Têt River catchment [28]. In effect, Ref. [85] showed that about 40% of suspended particulate matter was retained in the Vinça dam. However, the value is similar to those observed in larger rivers such as the Rhône ($\sim$15%, [86]). Nevertheless, it is also possible that this estimate of suspended sand transport has been slightly underestimated. Indeed, its estimation is based on the amount of sand collected on the surface, but as this sediment has a less homogeneous vertical distribution than the fine particles, this may lead to an underestimation of the suspended sand flux. Therefore, the sand contained in the SPM represents the deposition of 16% of the subaqueous delta (14,700 $\pm$ 400 m$^3$). With respect to bedload transport, its participation was estimated at 14% (12,400 $\pm$ 1800 m$^3$). Despite these various hypotheses and their uncertainties, the sediment balance of these deposits was not in equilibrium (20% deficit). We suggest that the remobilization of coastal sediments during the storm event, and their transport northwards by the action of the longshore drift and S-E waves, played a role. Although the cores suggest that resuspension is only observed at a depth of 10 m (as determined by the radioisotope $^{210}$Pb, Figure 9), the offshore displacement of the bars does not allow us to put forward an onshore contribution from this resuspension. Similarly, the accumulation to the south of the mouth (updrift coast) suggests a sediment trapping associated with the hydraulic barrier induced by the river jet during the flood peak. The contribution of the updrift coast sediment transported by the littoral drift to the subaqueous delta can then be considered to be negligible. One other hypothesis is the erosion of the channel upstream of the survey area. In this study, it was only possible to quantify the channel erosion for the first 100 m, representing 9% of the subaqueous delta deposited, and an extra contribution from the channel is probable. When the river mouth is closed, an accumulation in the channel is observed, and this accumulation probably affects more than the first 100 m. This latter hypothesis is related to the sedimentation of fine particles close to the river mouth (in the subaqueous delta). The presence of mostly fine sediments within the layer of flood deposits observable on cores St10 and St20 (determined by the radioisotope $^7$Be, Figure 9) confirms that a large proportion of fine sediments was exported offshore, unlike the sandy sediments, which were deposited in the subaqueous delta. However, a significant amount of fine sediment could have been trapped in the subaqueous delta by flocculation, increasing the total volume, as was observed in the Rhône delta in [87,88]. These last two hypotheses will be investigated in future work.

## 6. Conclusions

This study shows, for the first time, the monitoring of an extreme concomitant event that has been paid relatively little attention in the literature. It was observed that hydrodynamic processes were involved in the small temporal shift, with marine processes intervening first. These processes participated in weakening the sand spits and initiating breaching. The exceptional impact of the subsequent flooding helped to amplify the breaching phenomenon, leading to the complete destruction of the mouth. Thus, the concomitance of storm and flood during the Gloria event amplified the morphodynamic impact and led to a different result than would have been observed if either of these events had acted on the system independently. This shows the importance of studying these concomitant events in the context of climate change, and their potential increase in frequency and intensity in the near future. Moreover, our study highlights the relationship between the destruction of the mouth and the fluvial inputs leading to the creation of a large post-Gloria subaqueous delta. Even considering the uncertainty in our estimates, half of the inputs originated from the destruction of the river mouth, while only 30% were caused by the river flood. It is

important to note that the river contributes only a small part to the subaqueous delta in extreme flooding, compared to smaller events. The location of rainfall, particularly in catchment areas controlled by dams, is therefore a major consideration that can strongly affect the supply of sediment to the coast. The totality of the inputs that take part in the creation of this subaqueous delta could not be explained, but in view of the different avenues put forward, further work should be carried out to explore them in greater depth.

**Author Contributions:** Conceptualization, methodology, and writing, F.M., Y.B., N.R. and F.B.; analyses, F.M. and Y.B.; funding acquisition, Y.B. and F.B. All authors have read and agreed to the published version of the manuscript.

**Funding:** The corresponding author is funded through a PhD grant of French Occitanie Region and Bureau de Recherches Géologiques et Minières (BRGM). This work was supported by the DEM'EAUX project funded by the regional council Occitania and the FEDER.

**Institutional Review Board Statement:** Not applicable.

**Informed Consent Statement:** Not applicable.

**Data Availability Statement:** All data acquired in the framework of the DEM'EAUX project are available at the following address: https://demeauxroussillon.follow.solutions/.

**Acknowledgments:** We thank Météo France for supplying meteorological data, the DREAL Occitanie for hydrologic data and CEREMA/DREAL Occitanie/Observatoire Océanologique de Banyuls for wave data. We thank P. Feyssat and O. Raynal for the acquisition of post-Gloria bathymetry and N. Valentini for the X-Beach modeling. Thanks to the three anonymous reviewers, and the journal editor for their constructive and thoughtful comments and suggestions.

**Conflicts of Interest:** The authors declare no conflict of interest.

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
