# Peer review of "Assessing the Role of Extreme Mediterranean Events on Coastal River Outlet Dynamics"

_water, doi:10.3390/w14162463_

Round 1
Reviewer 1 Report
The authors report the monitoring of an extreme concomitant flood-storm event, which is rarely found in the literature. I have gone through the manuscript and paid attention to the soundness of the conclusions and the rigour of the methods. The methodology is complex involving hydrodynamic, topographic and bathymetric measurements, cores, as well as modelling, and it is therefore completely satisfactory.
The marine storm that occurred first appeared to weaken the river mouth spit which was later destroyed by the extreme flood, and a conceptual model is proposed. The authors report the deposition of a large subaqueous delta and estimate the river's direct contribution to this delta at 30%, lower than expected because rainfall occurred mostly behind a dam.
This study is important for the morphodynamics of river mouth areas and spits, for the understanding of transfers of sediment from rivers to coastal areas and beaches and for the impact of extreme events.
Considering the above, and how difficult it is generally to obtain data during extreme morpho-hydrodynamic events, I highly recommend this paper for publishing.
There are a few issues though that need to be tailored, as described below:
Although the flood and storm events partially overlap, there is a significant delay between peak storm and peak flood;
There is an efect on morphology because the spit didn’t had time to readjust to it’s previous morphology. Considering that he marine storm occured first (L23) then the flood, even though the events are linked, are they really concomitant? It’s more like quasi-concomitant.
L25-26: needs more clarity:
A large delta was deposited, of which we were able to estimate the contribution of the destruction of the mouth at 50%.
suggestion: .... the destruction of the barrier spit acted as a sediment source for the subaqueous large delta deposition amounting to 50% of the total volume ….
L46: can occur
The use of ‘delta’ throughout the paper is a bit unconventional. In coastal geomorphology, a delta is a deposit that has both subaerial and subaqueous parts (e.g. Nile delta). The deposit described as a ‘delta’ is purely subaqueous, therefore I suggest either specifying ‘subaqueous delta’ or subaqueous deposit. (such as in L72, L351, L490, L501….)
L112-114: Is there info about extreme percentiles? I suppose the 94 percentile is for Hs<1.5 but what are the percentile values for extreme events. Also, Hs mean could be nice to include for comparison to other sites.
L208: in what sense was the station destructed? Fortunately, data could be recovered.
L 226: this depth derivation in methods needs to come before, where depth from pressure was first mentioned.
L231-234: explain better how this calibration was done. Samples from what regions? What was the process/ exact steps taken
L248: could be nice to have a close-up view of the LiDAR data in one subplot.
L354: 1.5 m deep
L378: how was the bedload estimated?
L385: what is the density of the fluffy layer? was it obtained?
L389: present
L505: misplaced period.
L507: needs a closing parenthesis )
L524: very good observation regarding sediment trapping associated with the hydraulic groin effect.
L529: closed?
About the importance of first storm then flood:
L21:23 - The results suggest an amplification of the morphological impact of the events and a different morphogenic response than if each of the events had acted independently on the system.
+
L441-443: Even if the complete destruction of the mouth is the result of the impact of an exceptional flood, its influence remains secondary on spit breaching mechanism compared to marine processes.
→ I do not deny these statements, but it’s not clear to me how the data directly supports this. Still not clear how important is the barrier erosion pre-flood in the destruction of the barrier. Why wouldn’t the extreme flood just wash it away regardless of what happened before considering how thin and fragile is the spit/barrier, and that it was totally destroyed post-flood?
I would not say that the impact of the flood is secondary to spit breaching.
Even your stages (3-4) show that the barrier lithosome volume remains relatively constant (it's overwashed, it's breached but the sand is recycled and remains at the backbarrier).
The flood completely destroys the barrier and removes all the sediment. I would say it has a greater impact.
Would the barrier breach even without a storm? Most likely yes.
To have a definite answer we should compare a similar flood without the storm waves, and the same preexisting morphology, which is probably impossible at this site. Maybe there is some other site that is somehow sheltered from the waves of a hurricane but the watershed still receives intense precipitation. Also, a morphodynamic model testing scenarios could yield interesting results for the relative contribution of floods/storms in breaching this kind of barrier.
Figure 1: when is the image taken in plot b?
Figure 4: Hs line is not seen in Fig.4d. Needs to be added. Would have been nice to see a Hs during the review
Figure 6b is of low quality; I'm not sure it's like this because I have a temporary manuscript, but would be nice to have a higher-res photo for the event
Figure 7. Numbered areas not mentioned in the caption
I would like to see follow-up monitoring to understand the fate of the subaqueous ‘delta’ that got deposited post-flood.
My sincere salutations to the authors for this fine work,
Reviewer 2 Report
The article presents an extreme event in terms of morphodynamics at the mouth of a river, generated by the conjunction of a flood and an extreme marine episode. The event is extremely well documented and rich in information. This presentation will be of interest to the scientific community and to Water readers, and the data collected and presented in the article will constitute an extremely rich basis for numerical model applications, tests and comparisons. There is one point on which the authors could add a comment: their estimate of the transport of sand in suspension is based on the amount of sand collected at the surface (L 181), but the vertical distribution of sand is less homogeneous than that of fine particles, except in the case of extreme turbulence. It is therefore possible that the suspended sand flux has been underestimated based on surface sampling alone. I invite the authors to add a comment to this effect in their paper, if they share my view. I also suggest that they deposit their data in a data paper or open database, with a DOI, and reference it in the paper. But at this stage, the journal “Water” does not require this of its authors. Overall, I propose to accept the article, subject to a set of small detail corrections listed below. These are only minor corrections but will make the reading easier. I would like to congratulate the authors on their work.
Minor corrections:
L 39 : Please, add one parenthesis “(e.g. [7-9])”
L 40: please add one parenthesis “(e.g. [10-12])”
L 42 : “depend on the complex…
L 101 and following: please carefully check the units, km2 with 2 as superscript, m3 s-1 with 3 and -1 as superscripts and without any dot between m3 and s-1
L 101 and following: Hs with s as subscript
L 121: “is 150 m wide”
L 140: please replace “evolved” by “moved”
L 189: 50°C seems low. I know it is now accepted by the scientific community but I would suggest you to preferably dry the filters at 70°C in your future works, for a better accuracy (see https://www.somlit.fr/wp-content/uploads/2020/06/08-Protocole-national-MES-2020.pdf)
L 195, eq. 2: please correct “dicharge” into “discharge”
L 197: “a density of 1.6 T m-3”, the density is of 2.6 and the excess of density is 1.6 as compared to water, isn’t it?
L. 234, eq 3: please specify what is the unit of SPM
L 286 and following: please remove the dot between m and s-1 in the units.
L 295: please replace “is” by “was”
L 299: please replace” come” by “coming”
Fig. 4d: we can not see Hs. Please check the figure
L 331: 0 g L-1. I guess the minimum concentration was some mg L-1. Please check again the data series, it would be very surprising to have absolute clear water
L 353: please replace “the outlet is” by “the outlet was”
L 357: please replace “the mouth does” by “the mouth did”
L 397 and following: please check that 210 is written in superscript
L 400 and following: the unit of mass is kg and not Kg. Please correct it systematically. And the unit of activity is Bq kg-1 and not Bq.Kg-1 (without dot, -1 as superscript and kg instead of Kg). Please correct it all along the paper.
L 426: “overtopping was” instead of “is”
L 427: “induced” instead of “induce”
L 429 “contributed”
L 438: “completed”
L 496: after introducing Shaw et al 2018, please add “[75]”
L 507: please replace “T.an-1” in “t year-1” with year instead of an, no dot
L 516: please replace “estimate” by “estimated”
L 753: please correct “Rho# ne” into “Rhône” and add doi
Reviewer 3 Report
I can not see any areas where the paper needs additions before publication. But do have a couple of points for the author's consideration.
1. In terms of modeling, have the authors given any thoughts to using a code like DELFT 3D. A code that should have the capability of modeling the process studied here.
2. I very much like the conceptual modeling schematics—this adds significantly to the paper. To some degree this reminds me of the reduced complexity models developed by Andrew Ashton’s group at Woods Hole. For example
Lorenzo‐Trueba, Jorge, and Andrew D. Ashton. "Rollover, drowning, and discontinuous retreat: Distinct modes of barrier response to sea‐level rise arising from a simple morphodynamic model." Journal of Geophysical Research: Earth Surface 119.4 (2014): 779-801.
I wonder if there may be some benefit in referencing these works in the conceptual modeling section.
